# Flushing peripheral intravenous catheters: A scoping review

**Jiaxin Deng**[ID][1]*, **Orlaith Hernon**[ID][1], **Caitríona Duggan**[ID][2], **Leo R. Quinlan**[ID][3], **Zina Alfahl**[ID][4], **Peter J. Carr**[ID][1]

**1** School of Nursing and Midwifery, University of Galway, Galway, Ireland, **2** Department of Oncology, Portiuncula Hospital, Galway, Ireland, **3** Cellular Physiology Research Lab, Discipline of Physiology, School of Pharmacy and Medical Sciences, Galway, Ireland, **4** Discipline of Bacteriology, School of Medicine, University of Galway, Galway, Ireland

* J.deng3@universityofgalway.ie

## Abstract

### Background

Peripheral intravascular catheters (PIVCs) are indispensable vascular access devices in healthcare, facilitating the administration of intravenous therapies. Despite their vital role, PIVCs are frequently associated with complications such as occlusion, infection, and thrombosis, which contribute to catheter failure. Flushing catheters is one of the most common practices during PIVC maintenance, as it cleans the internal catheter lumen, ensuring patency and reducing the risk of complications. However, inconsistencies in flushing practices such as flushing technique, volume to use, frequency, and methods highlight a lack of consensus in the literature and clinical guidelines.

### Methods

Following JBI scoping review methodology, a comprehensive search was conducted across PubMed, Embase, Scopus, CINAHL, and grey literature sources. Studies were included if they focused on PIVC flushing techniques, flushing methods (speed, volume, frequencies, interval), or their impact on catheter-related outcomes. Data were charted using the PAGER (Patterns, Advances, Gaps, Evidence, Research recommendations) framework.

### Results

Of the 4539 initial studies retrieved, 39 met the inclusion criteria. Key findings reveal significant variability in flushing practices, with no consensus on optimal technique (continuous, intermittent, or pulsatile), volume (commonly 5–10 mL), or frequency (ranging from every 6 hours to every 24 hours). Pulsatile flushing showed promise in laboratory studies for reducing bacterial colonization and maintaining catheter

**Data availability statement:** All relevant data are within the paper and its Supporting information files.

**Funding:** The author(s) received no specific funding for this work.

**Competing interests:** The authors have declared that no competing interests exist.

patency but lacked consistent clinical evidence. Fluid dynamics studies on the flushing process suggested potential endothelial injury from high flushing velocities and the need for standardized practices.

## Conclusion

While some studies have investigated PIVC flushing, the existing research remains inconsistent, with a lack of clinical trials and mechanistic evidence on how flushing affects catheter patency, endothelial damage, and complication prevention.

---

### 1. Introduction

The peripheral intravenous catheter (PIVC) is a vital invasive device providing intravenous access for administering intravenous medications, solutions, and blood products [1]. PIVC insertion is the most frequently performed invasive procedure in hospitals, with up to 70% of patients requiring one [2]. with an estimated 2 billion PIVCs inserted worldwide annually [3].

Despite their widespread use and necessity, PIVCs are associated with various complications that can be broadly categorized as mechanical or physiological, which can lead to bacterial infectious complications. We suggest mechanical complications include device removal owing to securement failure, catheter dislodgement, and occlusion [4]. Physiological complications include thrombosis, extravasation, and phlebitis [4]. Additionally, interstitial edema may lead to the lifting of dressing edges, affecting catheter securement and increasing the risk of dislodgement and failure [5]. Infectious complications caused by bacteria may be caused locally at the insertion site or systemically if pathogens enter the bloodstream, which can result in thrombophlebitis in the presence of a thrombus [6].

PIVC complications are a burden to patients, clinical staff, and healthcare systems owing to catheter failure and repeat PIVC insertions [2]. The PIVC failure rate can range from 32% to 50%, with one study reporting a failure rate of 36.4%, and an overall incidence rate of 4.42 per 100 catheter days was reported [7,8]. Once the PIVC fails, there are delays in intravenous treatment delivery, patient dissatisfaction, and repeat insertion attempts are challenging as other veins are visibly depleted [9]. It is our contention that flushing the PIVC may contribute to these failure rates.

Drug incompatibilities, including calcium phosphate crystals, or lipid accumulation, can occur when medicines with different pH are infused, which increases the risk of catheter occlusion [10]. Clinical practice guidelines recommend flushing PIVCs as a critical intervention to effectively remove residual medications, maintain catheter patency, and minimize the risk of catheter occlusion [11–13]. The Infusion Therapy Standards of Practice recommend using a push-pause technique called pulsatile flushing with a 5 mL to 10 mL syringe of saline to flush PIVCs, ensuring the volume is twice that of the catheter system, both before and after drug administration [1]. This practice plays a role in assessing catheter function, identifying malfunction, and minimizing the risk of occlusion, thrombus, and the potential for catheter-related bloodstream infection [14].

The pulsatile flushing technique delivers the flushing solution in short bursts [15]. This technique induces turbulence within the catheter's internal lumen, reducing the time required for the deadhesion of solid deposits compared to flushing with a laminar flow [15]. However, the impact of flushing on the vein and blood components is poorly understood. This turbulence can be effective at cleaning solid deposits or biofilm fragments adhered to the internal catheter lumen. Whilst this action intends to clean the PIVC to ensure it is maintained and functional, this intervention often performed by nursing staff can result in physiological consequences as it creates high shear forces that can damage the vein [12]. The resulting stress and inflammation cause endothelial injury and may further encourage catheter failure owing to thrombus formation and interstitial edema [16]. The shear force created by the flush causes turbulence and recirculation at the catheter tip, which forces the platelets and blood cells to collide and may cause the catheter tip to be displaced or piston against the vein wall [17,18]. These potential adverse effects highlight the need for a deep understanding of the flushing practice and its implications, as current evidence remains insufficient to determine the optimal practices for PIVC maintenance [19]. Furthermore, this is a high-priority area of care for nursing as it is the discipline that performs the majority of care and maintenance of PIVCs.

This scoping review aims to systematically map the existing research on PIVC flushing techniques and methods, including the scope, sources of research, and types of evidence [20]. By identifying gaps in the current knowledge and any inconsistencies in practice, this review will provide a rationale for future research. The intention is to improve evidence-based guidelines for PIVC flushing and ultimately improve patient outcomes related to PIVC complications.

## 2. Scoping review objective and review questions

This scoping review aims to systematically map the existing research on PIVC flushing by assessing existing studies' scope, methodologies, and outcomes. The intention is to identify gaps in the literature to inform future research directions.

**Our review questions include:**

1. What is the body of the literature, including study methodologies, designs, and populations on PIVC flushing? What are the frequency distributions of contributing countries, authors, and clinical environments to the research output on this topic?

2. What descriptions of flushing techniques (e.g., pulsatile, continuous flushing) and flushing methods (e.g., flushing speed, flushing volume, flushing frequency, and flushing interval) are reported?

3. Does the literature examine how different catheter flushing methods contribute to catheter-related complications or catheter failure?

## 3. Method

Our published scoping review protocol adheres to the JBI scoping review methodological guidance [21]. The reporting of this review will follow the reporting guideline, the Preferred Reporting Items for Systematic Reviews and Meta-Analyses extension for Scoping Reviews (PRISMA-ScR) checklist [22]. The PAGER (Patterns, Advances, Gaps, Evidence for Practice and Research recommendations) framework will be used to help analyze and present the principle findings from the review [23]

## 4. Eligibility criteria

### 4.1. Participants, concept, and content

This study included healthcare professionals (HCP), such as nurses, medical doctors, paramedical HCPs, and vascular access specialists, who perform PIVC insertion, maintenance, and flushing processes, and laboratory professionals who conduct research in this field. The study population includes pediatric and adult patients, as well as animal models where applicable.

All types of peripherally inserted catheters, such as integrated, non-integrated, short, long, extended dwell, and midline catheters used for intravenous therapy (such as chemotherapy, medication administration, and parenteral nutrition), were included in the review. Midline catheters were not considered PIVCs but were included separately in the analysis if applicable. Studies focusing on flushing technique, flushing speed, flushing volume, flushing frequency, and flushing interval were included. Studies were excluded if they focused on the PIVC flushing or locking regimes or solutions (e.g., comparing the outcome difference between heparin and saline).

The context included clinical, laboratory, and simulated settings, including any environment used for training or research purposes, such as manikins, task trainers, and simulation technologies like virtual reality and augmented reality, as long as they relate to catheter flushing practices. In the clinical setting, both inpatient and outpatient settings were included.

### 4.2. Types of sources

The review included all study methodologies such as qualitative, quantitative, and mixed methods studies. The review included a variety of study designs, including non-experimental, experimental design, retrospective cohort studies, case-control studies, and simulation studies. We included evidence synthesis designs such as systematic reviews with and without meta-analysis and meta-synthesis and scoping reviews. Grey literature, such as conference papers and PhD thesis, were included.

### 4.3. Information sources

The search strategy was carried out following JBI's three-phase search strategy [24]. In the first phase, an initial limited search of the flushing of PIVCs was undertaken in PubMed and Embase to identify relevant articles on the topic of interest. The free text keywords and index terms noted from these relevant articles helped further devise our search terms and develop our search strategy. PJ, JD, OH, and CD contributed to developing search terms.

The search was conducted across several databases, including PubMed, Embase, Web of Science, Scopus, and Cumulative Index to Nursing and Allied Health Literature (CINAHL). Clinical trial registries such as the Australian and New Zealand Clinical Trial Registry (ANZCTR), European Union Clinical Trials Register (EUCTR), and ClinicalTrials.gov were also searched. Grey literature was identified by reviewing the first 10 pages of Google and Google Scholar using specific keyword combinations. The search strategy included free-text keywords and MeSH terms related to PIVC and flushing techniques (see Appendix I for detailed search terms and strategy). The search process was finished in October 2024.

### 4.4. Selection of sources of evidence

All identified literature was imported into Rayyan, and duplicates were removed. Two research team members (JD and OH) were involved in manually screening the literature retrieved from our dedicated search strategy. OH screened 10% of the literature to assess for agreement, and JD continued screening the remaining 90%. The third member (PC) independently used the ASreview tool for study screening, stopping after 100 consecutive non-relevant papers. To ensure a comprehensive review and minimize bias, the manual and ASreview screening results were compared. The research team members discussed discrepancies between the two methods to reach a consensus.

Full texts of potentially relevant studies were reviewed independently by one reviewer (JD). The second and third reviewers (OH and PC) also reviewed 10% of the studies to resolve discrepancies and ensure consistency.

### 4.5. Data charting process

A standardized data extraction form (Appendix II) was used to extract data, including author(s), year of publication, country of origin, study design, study settings, study subjects, flushing techniques, flushing speed, flushing volume, flushing

speed, and flushing frequency, flushing interval, and reported outcomes. One reviewer (JD) conducted the extraction work. We adopted a double-checking system with 10% of the studies, where second and third reviewers (OH and PC) verified the precision and comprehensiveness of the extracted data.

### 4.6. Analysis, presentation and discussion

Findings are presented using frequency counts and a narrative summary. Where appropriate, data are displayed using figures, tables, and graphs. Our discussion is divided into two sections: the first addresses the four scoping review questions, and the second highlights additional themes and findings identified during the review. The discussion further elaborates on both sections, guided by the PAGER framework [23].

### 4.7. Protocol amendments

Two amendments were made to our published scoping review protocol. Firstly, we limited the grey literature source to only include the first 10 pages of Google and Google Scholar with specific keyword combinations due to the large body of sources with low quality and irrelevant to the research questions. Secondly, the term PVC (peripheral venous catheter) in the scoping review protocol was replaced with PIVC (peripheral intravenous catheter) to align with standard terminology and better reflect the scope of our research.

## 5. Results

### 5.1. Studies inclusion

Our search strategy retrieved 4539 sources and 1418 duplicates were removed, leaving 3121 for title and abstract screening. After two authors' title and abstract screening, 71 studies were left for full-text screening. We subsequently conducted full-text screening, and then 39 studies met the final extracting requirements, as seen in PRISMA flow diagram (Fig 1) [25]. See Table 1 for a summary of information from the included studies.

### 5.2. Research design

We categorized the research designs into several types: 4 review studies (integrative and narrative reviews) [19,57–59], 6 intervention studies (including 5 randomized clinical trials) [26,27,29–31,60], 4 cross-sectional studies (descriptive and survey-based) [32–35], 11 observational studies (retrospective and prospective cohorts) [6,36–45], 9 experimental studies (including quasi-experimental and simulation studies) [15,17,46–52], 4 mixed-methods studies [53–56], and 1 guideline [14].

### 5.3. Research subject and setting

The research subjects are categorized as follows: 11 studies focused on healthcare professionals, specifically nurses and midwives, as the professional discipline performing the flushing practice [32–34,40–43,45,53–55]. 15 studies focused on hospitalized patients, including infants, newborns, and adults, assessing complications or outcomes associated with flushing [5,24,25,27–29,34–37,42,54–56]. 2 studies investigated hospital-wide flushing practices at an institutional level [35,59]. 10 studies were conducted in simulated or laboratory environments to evaluate flushing techniques under controlled conditions [15,17,43,46–49,51].

### 5.4. Flushing techniques

10 studies did not specify the flushing techniques, only vaguely described as manual flushing or flushing catheters [29,30,33,34,36,42,43,49,54,55]. 12 studies compared different flushing techniques, such as pulsatile flushing versus continuous flushing, or steady flow flushing versus pulsed flushing flow [6,15,26,27,37–39,44,47,48,50,51]. One RCT study

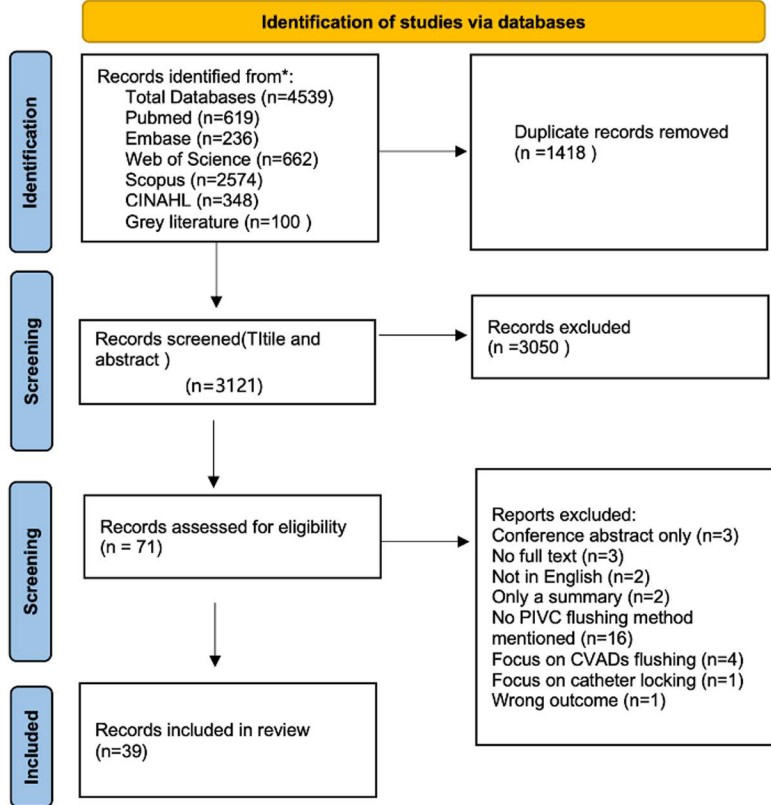

**Fig 1. PRISMA 2020 flow diagram.**

compared the effects of pulsatile flushing and continuous flushing. The result showed no statistical difference between pulsatile flushing and continuous flushing techniques regarding the time and type of PVC patency [27]. Other studies included intermittent flushing, continuous infusion, positive flush, and continuous and intermittent flushing combinations.

### 5.5. Flushing speed

The majority of the studies did not specify the flushing speed [6,15,26,29–37,39–43,45,49,54–56,60]. One simulation study mentioned a steady flow rate of 4 cm³/s with a syringe pump [15]. Another study described two different flushing methods: one group received 5 successive boluses of 1 mL each, completed in 0.5 seconds, while the other group received a single 5 mL bolus at a rate of 10 mL/min, with a flushing time of 30 seconds [48]. Other studies mentioned varying flow rates, such as 2.8, 6.8, 12.0, and 18.5 mL/s, or indicated that the flushing speed was "faster than recommended" [17,27,47,50]. Some studies focused on continuous infusion, reporting varying infusion rates such as 0.2 mL/h, 3 mL/h, 10 mL/h, 20 mL/h, and 40 mL/h [38,44,46]. With only one interventional study on flushing speed [48], there is an opportunity to develop an RCT comparing clinical outcomes of different flushing speeds.

### 5.6. Flushing volume

The flushing volumes are varied. 9 studies did not specify the amount [6,15,26,28,35,36,42,43,46]. The most mentioned volumes are 10 mL and 5 mL, often used alone or in comparison with other amounts like 2 mL or 3 mL. Some studies provided a range, such as 2–10 mL [6,26,28–35,38,39,44,46,47,54,55,59,60]. Two RCTs investigated the effect of flushing

**Table 1.** For a summary of information from the included 39 studies.

| Authors. | Research design | Research Subject | Research setting | Flushing technique | Flushing speed | Flushing volume | Flushing frequency | Flushing interval | Syringe Size | Outcome |
|---|---|---|---|---|---|---|---|---|---|---|
| Mali et al. 2022 [26] | Intervention study | Inpatients | a | IF VS Regular flushing | NS | NS | NS | NS | NS | 1, 2 |
| Hosseini et al. 2021 [27] | Intervention study | Inpatients | a | PF VS CF | PF:1 ml/s CF:1 ml/s | 5 ml | PF less than 1s delay | Less than 1s | NS | 2 |
| Keogh et al., 2020 [28] | Intervention study | Inpatients | a | PF | NS | NS | NS | NS | 10 ml pre-filled syringe | 3, 4 |
| Kledon et al., 2020 [29] | Intervention study | Inpatients | a | NS | NS | 1 Low frequency, low volume (q24h, 3 mL) 2 Low frequency, high volume (q24h, 10 mL) 3 High frequency, low volume (q6h, 3 mL) | NS | NS | NS | 3 |
| Keogh, et al, 2016 [30] | Intervention study | Inpatients | a | NS | NS | 1. High volume, high frequency (10 mL q6h) 2. High volume, low frequency (10 mL q24h) 3. Low volume, high frequency (3 mL q6h 4. Low volume, low frequency (3 mL q24h) | NS | NS | NS | 3, 5 |
| Schreiber et al, 2015 [31] | Intervention study | Inpatients | a | positive flush | NS | 3ml | q12h or q24h | NS | BD Posi-Flush XS syringes | 2, 4 |
| Parreira et al, 2020 [32] | Cross-section study | Nurses | b | CF and PF | NS | 2-10 ml | After PVC insertion, before, between, and after drug delivery | NS | 2,5,10 ml | 6 |
| Keogh, et al,2015 [33] | Cross-section study | Nurses | b | NS | NS | 2-10 mL, with 10 mL most common | q12h or q24h | NS | BD Posi-Flush XS syringes | 6 |
| Braga et al., 2018 [34] | Cross-section study | Inpatients and Nurses | c | NS (manual flushing) | NS | 3 ml, 5 ml, 10 ml | NS | NS | 3 ml, 5 ml, 10 ml syringe | 7 |
| Cabrero et al, 2005 [35] | Cross-section study | Hospital | b | IF and CF | NS | NS | NS | NS | NS | 6 |
| Tseng et al, 2022 [6] | Observation study | Inpatients | a | (hypertonic, isotonic and hypotonic) osmolarity continuous intravenous drip VS IF | NS | NS | NS | NS | NS | 4, 5 |
| Campbell et al., 2005 [36] | Observation study | Inpatients (ambulatory patients) | a | NS (manual flushing) | NS | NS | q8h, q24h | NS | NS | 4 |
| Flint et al., 2008 [37] | Observation study | Inpatients | a | CF (10% dextrose) VS IF with 2 mL 0.9% sodium every 6 hours | NS | 2 ml | q6h | NS | NS | 5 |

*(Continued)*

| Authors. | Research design | Research Subject | Research setting | Flushing technique | Flushing speed | Flushing volume | Flushing frequency | Flushing interval | Syringe Size | Outcome |
|---|---|---|---|---|---|---|---|---|---|---|
| Hoff et al., 2019 [38] | Observation study | Inpatients (infants) | a | CF: 0.2 mL/h VS IF: 5 mL before and 0.3 mL after the administration of intravenous medication | NS | IF:5 mL before and 0.3 mL | Before and after administration of intravenous medication | NS | NS | 2, 4, 8 |
| Stok et al., 2016 [39] | Observation study | Inpatients (infants) | a | Continuous infusion with 5% dextrose at 3 ml/h VS IF with 2 ml 0.9% saline six times daily | NS | IF: 2 mL | IF: q4h | NS | NS | 2, 4, 8, 9 |
| Ribeiro et al, 2023 [40] | Observation study | Nurses | d | continuous low-flow flushing and PF | NS | 5 mL and 10 ML | NS | NS | NS | 6 |
| Nunes et al., 2022 [41] | Observation study | Nurses | d | PF | NS | Around 5 ML | Before, between, and after medication administration | NS | 10 ml | 4 |
| Lee et al., 2021 [42] | Observation study | Health professionals | a | NS | NS | NS | NS | NS | Manually prepared syringe VS pre-filled syringe | 9, 10, 11, 12 |
| Keogh et al., 2014 [43] | Observation study | Nurses | e | NS | NS | NS | NS | NS | NS | 6, 9, 10 |
| Perez et al., 2012 [44] | Observation study | Inpatients (infants) | a | Continuous infusion (0.9% saline) VS IF (1 mL per day; if needed, antibiotics are flushed every 8h) | Not specified for IF CF: 2 ml/h | Continuous infusion: 2 ml/h; IF: 1 ml per day | IF: 1 ml | q24 or q8h | NS | 2 |
| Wotton et al., 2004 [45] | Observation study | Nurses | b | IF | NS | 2-10 ml | NS | NS | NS | 6 |
| Doyle et al., 2021 [46] | Experimental study | Lab study on catheters | e | Continuous infusion | KVO infusion:10 mL/h, 20 mL/h, and 40 mL/h | NS | NS | NS | NS | 2, 13, 14 |
| Okamura et al, 2003 [47] | Experimental study | Lab study on catheters | f | PF VS CF | CF: 10 mL/4, 7, or 10s PF: 10 mL/4, 7, or 10s with 0.2,0.4, 0.8s pause with 1 ml inject | 10 ml | 1 time | PF: 0.2,0.4, 0.8s | NS | 15 |
| Zhu et al., 2020 [17] | Experimental study | Lab study on catheters | e | PF with varied bolus volumes | 2.8, 6.8, 12.0, 18.5 mL/s. | 0.5, 1, 1.5, and 2.0 mL | Once | 0.5 and 0.4 s | NS | 16, 17 |

*(Continued)*

| Authors. | Research design | Research Subject | Research setting | Flushing technique | Flushing speed | Flushing volume | Flushing frequency | Flushing interval | Syringe Size | Outcome |
|---|---|---|---|---|---|---|---|---|---|---|
| Tong et al., 2019 [48] | Experimental study | Rabbit | f | PF, CF and control group | PF: 5 successive boluses, 1 mL flushed in 0.5 s each. CF: single 5 mL bolus (10 mL/min, flushing time is 30 s) | 5 ml | q8h | PF:0.4s | 5ml | 1, 2, 18 |
| Marques et al, 2019 [49] | Experimental study | Lab study on catheters | f | NS | NS | 10 ml | NS | NS | NS | 19 |
| Ferroni et al, 2014 [50] | Experimental study | Lab study on catheters | f | PF VS CF | PF: 1 ml/s CF: 1 ml/s | 10ml | q24h | NS | NS | 15 |
| Guiffant et al, 2011 [51] | Experimental study | Lab study on catheters | f | PF, Continuous infusion, CF | Varied | CF: 10 ml Continuous infusion: 500 ml PF:10 ml | q24h | Varied | NS | 19 |
| Chittick et al., 2010 [52] | Experimental study | Lab study on catheters | f | PF | PF: injecting 0.5 mL of saline 20 times over 10 seconds | 10ml | Once | NS | Syringe pump | 15 |
| Vigier et al., 2005 [15] | Experimental study | Lab study on catheters | f | steady flow vs. pulsed flow | NS | NS | NS | NS | NS | 16 |
| Ribeiro et al, 2023 [53] | mix-methods study | Nurses | b | Varied | Varied | Varied | Varied | NS | NS | 6 |
| Santos et al., 2022 [54] | mix-methods study | Inpatients and Nurses | c | NS | NS | Around 5 ML | Before, between, and after medication administration | NS | 10 ml | 4 |
| Norton et al., 2019 [55] | mix-methods study | Nurses | c | NS | NS | 2-10 mL, with 10 mL most frequent | NS | NS | NS | 6 |
| Varal-akshmi et al, 2018 [56] | mix-methods study | Inpatients | a | IF | NS | 2 ml | q12h VS None | NS | NS | 2 |
| Cullinane et al., 2019 [14] | guideline | – | – | – | – | – | – | – | – | – |
| Ribeiro et al, 2022 [57] | review | – | – | – | – | – | – | – | – | – |
| Hawthorn et al., 2019 [19] | review | – | – | – | – | – | – | – | – | – |

*(Continued)*

**Table 1.** (Continued)

| Authors. | Research design | Research Subject | Research setting | Flushing technique | Flushing speed | Flushing volume | Flushing frequency | Flushing interval | Syringe Size | Outcome |
|---|---|---|---|---|---|---|---|---|---|---|
| Flint et al., 2005 [58] | review | – | – | – | – | – | – | – | – | – |
| Fernandez et al, 2003 [59] | review | – | – | – | – | – | – | – | – | – |

**Research Subject: A** inpatients **B** Nurses; **C** Nurses and inpatients; **D** hospital; **E** Health professionals; **F** lab study on catheters; **G** Rabbit.

**Research Settings: a** Clinical environment; **b** Survey or questionnaires; **c** Mixed settings; **d** Observation in real clinical; **e** Simulation setting.

**Research Design: I** intervention study; **II** cross-sectional study; **III** observational study; **IV** experimental study; **v** mixed-methods study; **vi** guideline; **vii** review.

**Outcomes: 1** incidence of phlebitis; **2** Maintenance of catheter patency; **3** PIVC failure; 4 PIVC complications; **5** PIVC dwell time; **6** flushing practice evaluation (adherence to guideline practice); **7** incidence of catheter obstruction; **8** material cost; **9** time cost; **10** Risk of contamination; **11** Needlestick injuries; **12** Medication errors; **13** blood stasis; **14** shear stress on the vein; **15** bacterial growth/colonization; **16** flow characteristics; **17** mechanical behaviors;

[18] histopathological changes; **19** protein removal efficiency/ albumin recovered.

**Abbreviation: NS**: Not specified; **PF**: Pulsatile flushing; **IF**: Intermittent flushing; **CF**: Continuous flushing; **Versus:** VS.

volume in combination with frequency [29,30]. Both compared low-volume (3 mL) versus high-volume (10 mL) flushing. The results showed no significant difference in PVC failure rates between the low- and high-volume groups. Overall, while 10 mL and 5 mL are common, there is variability and some lack specification.

### 5.7. Flushing frequency

We identified 13 studies that report scheduling of flushing is varied with no definite flushing frequency [5,13,24,26,32,33,38,40,41,43,44,47,53]. Further, 20 studies noted variability based on clinical circumstances and requirements, such as more frequent flushing during night shifts, after PVC insertion, before, between, and after drug delivery, or as needed [25,27–31,34,37,39,42,45,46,48–52,54]. Two RCTs also assessed flushing frequency, comparing low-frequency flushing (every 24 hours) with high-frequency flushing (every 6 hours). The findings indicated no significant reduction in PVC failure rates with more frequent flushing [29,30]. Common specific flushing frequencies were included in n = studies q24h(every 24 hours), q6h(every 6 hours), q8h(every 8 hours), and q12h(every 12 hours), often with medication administration [29–31,36,39,44,47,48,50–52,60]. Overall, the included studies have a mix of specific and variable frequencies.

### 5.8. Syringe used for flushing

The results regarding flushing syringe sizes, types, and prepared methods varied. A substantial portion (n = 20) of the included studies did not specify the syringe type, sizes, and prepared methods. Specific syringe sizes ranged from 2 ml to 10 ml and included 2 ml, 3 ml,5 ml, and 10 ml,. Certain studies highlighted brand-specific syringes, such as the BD Posi-Flush and BD PosiFlush XS syringes [27,43], and one study reported using a syringe pump [48].

### 5.9. Study outcome assessment for included studies

The included studies evaluated how various factors related to PIVC flushing influence catheter-related complications, describes as infection, infiltration, extravasation, occlusion, phlebitis, and thrombosis, as well as catheter failure. Several studies investigated the practitioner's performing the catheter flushing in adherence to established guidelines (flushing frequency, volume, speed) [33,35,40]. A subset of studies also examined mechanical behaviors, flow characteristics of

flushing techniques, and the impact of residual liquids on bacterial growth [47,50,61]. These outcome assessments collectively highlight the importance of safety, effectiveness, and protocol adherence in PIVC management.

## 6. Discussion

This scoping review aimed to comprehensively map the current state of PIVC flushing practices, focusing on the details of PIVC flushing process, revealing the diversity and complexity of research in this field. Our analysis included various geographical locations, study designs, study subjects, and flushing details, providing a broad perspective on PIVC flushing techniques and methods. The findings highlight both the progress made in understanding PIVC flushing and the challenges in establishing a standardized PIVC flushing practice. Using the PAGER framework,[19] we discuss the main scoping review findings, which includes patterns, advances, gaps, evidence for practice, and research recommendations for PIVC flushing, as seen in Table 2.

### 6.1. Variation in PIVC flushing practice

The lack of a standardized protocol for PIVC flushing practices has led to significant variation across different regions and healthcare settings. Several factors, including differences in educational training, availability of medical resources, and staffing levels, influence this variation in practice [62]. Additionally, healthcare professionals may have a limited understanding of the importance of proper PIVC management, which can contribute to inconsistent practices [63,64]. Implementation studies are limited, with few studies exploring how research on PIVC flushing can be effectively translated into clinical workflows. One study evaluated errors in nursing teams' flushing practices and developed a flushing prototype—a structured care guide to promote good flushing practices in intensive care units [53]. A single-center, stepped-wedge, cluster-randomized trial demonstrated the impact of a multifaceted intervention focused on PIVC maintenance [28]. The intervention, which included education on practice guidelines and the use of manufacturer-prepared pre-filled flush syringes, resulted in a significant reduction in PIVC failure rates—30% in the control group vs. 22% in the intervention group (risk difference −8%, 95% CI −14

**Table 2. PAGER framework.**

| Pattern | Advances | Gaps | Evidence for Practice | Research Recommendations |
|---|---|---|---|---|
| Variation in PIVC flushing practice | Research covers multiple regions | Lack of multicenter studies representing diverse countries | Current practices vary widely across hospitals, indicating a need for standardized protocols to ensure consistent patient care. | Collaborative, multicenter studies are needed to explore effective flushing methods across different healthcare environments |
| The diversity in PIVC flushing techniques | Explored different techniques like pulsatile, continuous, and intermittent flushing in clinical outcomes | Lack of detailed and precise records of different flushing techniques, such as Lack of detailed and precise records specifying whether push-pause or continuous flushing techniques were used. | Current guideline recommend flushing techniques, but there is no clear evidence on the most effective method. | Conduct clinical trials to compare various techniques and conduct evidence-based practices |
| Physiological impact and safety of different flushing techniques | Intervention studies have explored different flushing techniques and methods, analyzing their varying outcomes. | There is no consensus on optimal volume and frequency, and many studies lack precise flushing methods and details | Larger-scale intervention studies and flushing physiological impact basic experiments are needed to establish optimal practices. | Further clinical trials and basic experiments from physiological safety aspect to identify the best combination of volume and frequency in varied settings |
| Lack of research on the mechanisms of PIVC flushing | Some studies suggest that flushing methods (e.g., pulsatile) may prevent complications like infection and thrombosis | Very few studies explore how different flushing techniques affect catheter patency or prevent complications such as damage to the endothelium. | More clinical relevant research on the mechanisms of PIVC flushing is needed to bridge the gap between experimental findings and real-world practice. | Future research should focus on understanding the underlying mechanisms of PIVC flushing to guide the development of evidence-based practices |

to −1, p = 0.032). Importantly, the intervention also reduced total costs without any serious adverse events. These implementation studies emphasize the need to bridge the gap between research and practice, offering an opportunity for translational and implementation science to develop practical strategies that enhance guideline adherence.

### 6.2. The diversity in PIVC flushing techniques, speed, volume, frequency, and interval

One of the purposes for carrying out a scoping review is to identify the number of randomized controlled trials to justify the need for a full systematic review and meta-analysis [20]. In this scoping review, we included various clinical study designs, including RCT (n = 5), cohort studies (n = 4), and case-control study (n = 1). Three of the five RCT studies aimed to compare the high PIVC flushing volumes(3 ml or 10 ml) and frequencies(every 6 hours or every 24 hours) to evaluate the impact on catheter failure and complication rates [29–31]. However, significant variability existed across these studies regarding flushing techniques, volumes, frequencies, patient populations, and other factors such as PIVC gauge and insertion site. This heterogeneity causes considerable confounding, limiting the ability to draw definitive conclusions about the optimal flushing strategies. Additionally, methodological differences—such as variations in patient populations (e.g., adult vs pediatric), treatment types (e.g., antibiotics or parenteral nutrition) definitions of outcomes (e.g., catheter failure rates vs dwell time). Given these limitations, conducting a systematic review with meta-analysis would be beneficial to quantitatively synthesize the available evidence and to clarify the relationship between specific flushing parameters and catheter outcomes. Additional rigorous clinical trials are essential to determine the optimal flushing techniques, volumes, and frequencies, particularly across different patient populations and clinical treatment types.

### 6.3. Physiological impact and safety of different flushing techniques

The included studies include several PIVC flushing techniques, including continuous flushing, intermittent flushing, and pulsatile flushing, representing current clinical flushing techniques. Continuous flushing provides a steady flow of solution with no interval, while intermittent flushing is used at set intervals to maintain catheter patency [1]. Pulsatile flushing, which involves short bursts of solution with pauses, is recommended to enhance debris clearance from the catheter [1,65]. However, current practice guidelines advocate using pulsatile flushing primarily based on experts' opinions with limited evidence [12]. Ferroni et al. demonstrated that pulsatile flushing significantly reduced bacterial colonization, particularly for Staphylococcus aureus, compared to continuous flushing [50]. Boord et al. found that pulsatile flushing is better for maintaining catheter patency and clearing solid deposits from the catheter walls [12]. These findings support the theoretical benefits of pulsatile flushing; however, this evidence comes from in vitro studies, limiting the generalizability of the results to clinical practice. However, an included RCT study compared the effects of continuous flushing on PIVC and found no difference between continuous flushing and pulsatile flushing regarding the time and type of PIVC patency [27]. Another prospective study compared the duration of PIVC patency between continuous flushing and intermittent flushing and found that intermittent flushing resulted in a significantly longer cannula patency duration than continuous infusion (geometric mean 47.1 vs. 35.4 h, P = 0.041). The incidence of extravasation was higher with continuous infusion (68.9% vs. 43.2%; P = 0.001), while occlusion was more common with intermittent flushing (28.4% vs. 6.6%; P = 0.002) [44]. The pulsatile flushing shows theoretical benefits in laboratory settings, its clinical advantages remain uncertain. These mixed results suggest that the optimal flushing technique may vary depending on the clinical context and patient population. Further research is needed to establish standardized guidelines for these variations and ensure the best clinical outcomes.

### 6.4. Lack of research on the mechanisms of PIVC flushing

The understanding of the mechanisms behind PIVC flushing is limited, and few studies have explored fluid dynamics in the PIVC flushing process. Fluid dynamics in the context of PVC flushing refers to the study of fluid flow behavior and forces within the catheter during the flushing process. This includes how the flushing saline interacts with the catheter walls, generates turbulence and creates shear forces to remove the solid deposits in the internal catheter [17]. The studies by Vigier et al. and Ferroni et al. examined pulsatile flushing but used different methods [15,50]. Vigier et al., used a transparent rectangular duct

(10 cm × 1.5 cm × 0.2 cm) connected to a pump to create controlled steady or unsteady flow. A thin layer of solid deposit was formed at the bottom by evaporating a mineral powder suspension in water, simulating catheter deposits. It took 70 seconds to initiate deposit removal under steady flow conditions (4 mL/s), whereas applying an intermittent flow, with the velocity doubling at 2-second intervals, reduced this time to 25 seconds. Ferroni et al., used polyurethane catheters contaminated with a controlled concentration of Staphylococcus aureus. Two flushing techniques controlled by a pump were compared: pulsative flushing (ten 1 mL boluses of saline) and continuous flushing (a single 10 mL bolus). Catheters flushed with the pulsative method had a significantly lower median bacterial count of 524 CFU/mL compared to 1,616 CFU/mL observed in catheters flushed with the continuous method, demonstrating the higher effectiveness of the pulsative technique in reducing bacterial contamination. However, the study by Zhu et al., investigated the mechanisms of pulsatile flushing using various bolus volumes and found that higher bolus volumes (1.5 mL and 2.0 mL) within 0.5 seconds, generating peak shear rates up to 10,000 $s^{-1}$ can significantly increase the risk of mechanical endothelial injury [17]. The range for shear rates in *vivo* for the vein is 20–200 $s^{-1}$ [65]. However, shear rates over 10,000 $s^{-1}$ near the vessel wall induce platelet adhesion to thrombogenic surfaces [66,67]. Shear rates over 10,000 $s^{-1}$ near the vessel wall induce platelet adhesion to thrombogenic surfaces. This process is mediated by the von Willebrand factor (vWF), which undergoes conformational changes under high shear, exposing binding sites that facilitate platelet aggregation. Elevated shear rates can also promote platelet activation and secretion of procoagulant substances, enhancing thrombus formation [68,69]. The experimental setup allowed for precise control over variables such as flow rate, bolus volume, and interval timing, allowing for accurate results regarding the effectiveness of flushing [40]. Educational materials/ clinical guidelines typically describe pulsatile flushing as a one-second flush followed by a one-second pause with no detailed description of flow rate, bolus volume, and interval timing [1]. Furthermore, there is a lack of detailed clinical data on actual flushing practices, making it difficult to determine whether the potential benefits of pulsatile flushing are superior to continuous infusion, such as keeping vein open approaches or manual flushing at a prescribed approach, outweigh the risks of endothelial injury. No meta-analysis has been conducted to evaluate the efficiency or efficacy of different flushing methods in clinical practice. Conducting a meta-analysis of existing studies is justified, as is the design of clinical simulation studies, conducting lab-based experiments measuring endothelia injury with flushing. Such approaches are essential to establishing robust evidence for PIVC flushing practices.

## 7. Conclusion

There are diverse and often inconsistent practices surrounding PIVC flushing. Various flushing techniques, such as continuous, intermittent, and pulsatile flushing techniques, with no consensus on the optimal volume, frequency, or technique to maintain PIVC patency and reduce PIVC complications. Despite the theoretical benefits of specific methods like pulsatile flushing, the evidence remains limited, and clinical data are rare. To address these challenges, further collaborative transdisciplinary research is needed to explore and understand the mechanism of flushing a PIVC in clinical practice.

## Supporting information

**S1 Table. Search terms and synonyms used for the literature search.**
(DOCX)

**S2 Table. Data extraction form for included studies.**
(DOCX)

## Author contributions

**Conceptualization:** Jiaxin Deng, Orlaith Hernon, Caitríona Duggan, Leo R Quinlan, Peter J Carr.

**Data curation:** Jiaxin Deng.

**Formal analysis:** Orlaith Hernon.

**Methodology:** Jiaxin Deng, Caitríona Duggan, Peter J Carr.

**Software:** Jiaxin Deng.

**Supervision:** Leo R Quinlan, Peter J Carr.

**Writing – original draft:** Jiaxin Deng, Caitríona Duggan, Peter J Carr.

**Writing – review & editing:** Orlaith Hernon, Leo R Quinlan, Zina Alfahl, Peter J Carr.

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
