## [Decision Letter · Decision Letter 0]

9 Jun 2025

Dear Dr. Deng,

Thank you for submitting your manuscript to PLOS ONE. After careful consideration, we feel that it has merit but does not fully meet PLOS ONE’s publication criteria as it currently stands. Therefore, we invite you to submit a revised version of the manuscript that addresses the points raised during the review process.

Please address the reviewers' comments as mentioned.

We look forward to receiving your revised manuscript.

Kind regards,

Erik Su

Academic Editor

PLOS ONE

2. In the online submission form, you indicated that your data will be submitted to a repository upon acceptance.  We strongly recommend all authors deposit their data before acceptance, as the process can be lengthy and hold up publication timelines. Please note that, though access restrictions are acceptable now, your entire minimal  dataset will need to be made freely accessible if your manuscript is accepted for publication. This policy applies to all data except where public deposition would breach compliance with the protocol approved by your research ethics board. If you are unable to adhere to our open data policy, please kindly revise your statement to explain your reasoning and we will seek the editor's input on an exemption. 

Additional Editor Comments (if provided):

Reviewers' comments:

Reviewer's Responses to Questions

**Comments to the Author**

1. Is the manuscript technically sound, and do the data support the conclusions?

Reviewer #1: Yes

Reviewer #2: Yes

2. Has the statistical analysis been performed appropriately and rigorously?

Reviewer #1: Yes

Reviewer #2: N/A

3. Have the authors made all data underlying the findings in their manuscript fully available?

Reviewer #1: Yes

Reviewer #2: Yes

4. Is the manuscript presented in an intelligible fashion and written in standard English?

Reviewer #1: Yes

Reviewer #2: Yes

Reviewer #1: Thank you all for your work. It is clear you spent a great deal of time and effort putting together a well thought out review.

I do not have any major revisions for you to complete only small adjustments below:

- In your introduction near line 110 it appears the first sentence of the new paragraph is a repeat of the description/ definition of pulsatile flushing that is mentioned up above. Nothing major it just reads as if you are defining it twice.

- For table 2- I did not see outcome 17 listed as being used in the table. Perhaps it was just not assigned to a paper or you could just remove it as an outcome.

- On section 5.5 on flushing speeds it may be worth considering converting the 4 cm3/sec to ml/sec to have consistency in the labeling of units throughout the manuscript. Again, near sentence at number line 478.

- For table 3 for the last row under the column Gaps- would reword the sentence to read “… patency or prevent complications or damage to the endothelium”. Essentially removing the word damage again at the end.

- Around line number 490- For the data on shear rates- I could not see where ref 67 Dunkley et. al states specific shear rates related to platelet aggregation. And I didn’t see mention of them by Ref 65 Nicholson. Perhaps the reference numbers got shifted a little.

Overall, this is a great review of the literature and a good summary of what is available to us.

Reviewer #2: Very interesting topic reviewed, especially the section on fluid dynamics.

Only point is the possible grammatical error of line 78 and PVC used without the explanation of what it stands for. Was this intended to be PIVC? PVC is later used and explained in line 251.

**Do you want your identity to be public for this peer review?** For information about this choice, including consent withdrawal, please see our Privacy Policy

Reviewer #1: No

Reviewer #2: No

---

## [Author Response · Author response to Decision Letter 1]

17 Jun 2025

Reviewer #1 comments & responses

Comment: In your introduction near line 110 it appears the first sentence of the new paragraph is a repeat of the description/ definition of pulsatile flushing that is mentioned up above.

Response: Thanks for your comment. We have revised the sentence in Line 128.

Comment: For table 2 - I did not see outcome 17 listed as being used in the table. Perhaps it was just not assigned to a paper or you could just remove it as an outcome.

Response: Thank you for your comment. Outcome 17 was indeed used in Table 2 and assigned to the study by Zhu et al. (2020) on Page 19.

Comment: On section 5.5 on flushing speeds it may be worth considering converting the 4 cm³/sec to ml/sec to have consistency in the labeling of units throughout the manuscript.

Response: Thank you for your comment. We have revised the unit from “4cm³” to “4mL/s” in section 5.5 and Line 514.

Comment: For table 3, for the last row under the column Gaps — would reword the sentence to read “… patency or prevent complications or damage to the endothelium”. Essentially removing the word damage again at the end.

Response: Thank you for your comment. We have revised the sentence as seen in Table 3.

Comment: Around line number 490 — For the data on shear rates — I could not see where ref 67 Dunkley et al. states specific shear rates related to platelet aggregation. And I didn’t see mention of them by Ref 65 Nicholson. Perhaps the reference numbers got shifted a little.

Response: Thank you for your helpful comment. We have reviewed the references and identified that the original References 65 and 67 did not adequately support the statements regarding physiological and pathological shear rates and their impact on platelet aggregation. We have replaced Reference 65 with Shi et al. (2015), which provides a detailed explanation of shear rate ranges in vein. Reference 67 has been updated to Casa et al. (2016), which reports on the effects of different shear rates on platelet thrombi formation and detachment. Both references have been appropriately cited in the revised text, as seen in Line 791 and Line 796.

Reviewer #2 comments & responses

Comment: Only point is the possible grammatical error of line 78 and PVC used without the explanation of what it stands for. Was this intended to be PIVC? PVC is later used and explained in line 251.

Response: Thanks for your comment. It was a typo in Line 78 — should have been PIVC (peripheral intravenous catheter). We have corrected this in Line 96.

---

## [Decision Letter · Decision Letter 1]

27 Jul 2025

Flushing Peripheral Intravenous Catheters: A Scoping Review

PONE-D-25-22430R1

Dear Dr. Deng,

We’re pleased to inform you that your manuscript has been judged scientifically suitable for publication and will be formally accepted for publication once it meets all outstanding technical requirements.

Kind regards,

Erik Su

Academic Editor

PLOS ONE

Additional Editor Comments (optional):

Reviewers' comments:

Reviewer's Responses to Questions

**Comments to the Author**

Reviewer #1: All comments have been addressed

Reviewer #2: All comments have been addressed

2. Is the manuscript technically sound, and do the data support the conclusions?

Reviewer #1: Yes

Reviewer #2: Yes

3. Has the statistical analysis been performed appropriately and rigorously?

Reviewer #1: Yes

Reviewer #2: Yes

4. Have the authors made all data underlying the findings in their manuscript fully available?

Reviewer #1: Yes

Reviewer #2: Yes

5. Is the manuscript presented in an intelligible fashion and written in standard English?

Reviewer #1: Yes

Reviewer #2: Yes

Reviewer #1: No additional comments. Thank you for your updates.

Reviewer #2: (No Response)

**Do you want your identity to be public for this peer review?** For information about this choice, including consent withdrawal, please see our Privacy Policy

Reviewer #1: **Yes: ** Mark D. Weber MSN, RN, CRNP-AC, FCCM

Reviewer #2: No

---

## [Editor Report · Acceptance letter]

PONE-D-25-22430R1

PLOS ONE

Dear Dr. Deng,

I'm pleased to inform you that your manuscript has been deemed suitable for publication in PLOS ONE. Congratulations! Your manuscript is now being handed over to our production team.

Kind regards,

on behalf of

Dr. Erik Su

Academic Editor

PLOS ONE